# The Effects of Differentiated Organic Fertilization on Tomato Production and Phenolic Content in Traditional and High-Yielding Varieties

**DOI:** 10.3390/antiox11112127

**Published:** 2022-10-28

**Authors:** Johana González-Coria, Julián Lozano-Castellón, Carolina Jaime-Rodríguez, Alexandra Olmo-Cunillera, Emily P. Laveriano-Santos, Maria Pérez, Rosa Mª Lamuela-Raventós, Jordi Puig, Anna Vallverdú-Queralt, Joan Romanyà

**Affiliations:** 1Department of Biology, Health and the Environment, Faculty of Pharmacy and Food Sciences, University of Barcelona, 08028 Barcelona, Spain; 2Institute of Nutrition and Food Safety (INSA-UB), University of Barcelona, 08028 Barcelona, Spain; 3Department of Nutrition, Food Science and Gastronomy XIA, Faculty of Pharmacy and Food Sciences, University of Barcelona, 08028 Barcelona, Spain; 4CIBER Physiopathology of Obesity and Nutrition (CIBEROBN), Institute of Health Carlos III, 28029 Madrid, Spain; 5Laboratory of Organic Chemistry, Faculty of Pharmacy and Food Sciences, University of Barcelona, 08028 Barcelona, Spain; 6L’Espigall, 08480 l’Ametlla del Vallès, Spain

**Keywords:** organic agriculture, soil, foliar nutrients, abiotic factors, polyphenols, antioxidants, fruit quality, health, *Solanum lycopersicum*

## Abstract

The challenge of sustainable agriculture is to increase yields and obtain higher quality products. Increased antioxidant compounds such as polyphenols in harvest products may be an added value for sustainable agriculture. The aim of the present study was to investigate whether three organic fertilization treatments with different levels of carbon and nitrogen, i.e., N-rich, N-rich+C, and N-poor+C, affected the phenolic content of different tomato varieties. The examined parameters were productivity, plant nutritional status, δ^13^C, and tomato phenolic content as an indication of the antioxidant capacity. The best production was obtained with ‘Cornabel’, a high-yielding Pebroter variety. The total phenolic content was highest in the traditional ‘Cuban Pepper’ variety regardless of treatment, while naringenin levels were high in all the Pebroter varieties. In N-poor+C fertilized plants, a lower N-NO_3_ content in leaves was correlated with higher levels of total polyphenols in the fruit. The high-water stress suffered by Montserrat varieties coincided with a low total phenolic content in the tomatoes. In conclusion, organic fertilization with reduced N did not influence the tomato yield but positively affected phenolic compound levels in varieties less sensitive to water stress.

## 1. Introduction

The tomato (*Solanum lycopersicum)* is one of the most popular horticultural crops worldwide and is a key component of the Mediterranean dietary pattern. Global tomato production is around 180 million tons per annum, more than a quarter of which is processed to make sauces and other derivatives [1]. Previous work has shown that differences in soil fertility can change the composition of tomatoes and can have significant effects on their organoleptic properties and health-promoting components [2]. The tomato is an important source of substances with known beneficial effects on health which are considered natural antioxidant compounds such as lycopene, ascorbic acid, tocopherols, polyphenols, pro-vitamin A, and vitamin C [3,4]. Indeed, tomato consumption has been associated with a reduced risk of cancer, inflammatory processes, coronary heart disease, hypertension, obesity, diabetes, and other cardiovascular conditions [5].

In addition to increasing soil organic matter, the application of organic amendments is a way of replacing nutrient exports from previous crops [5]. The application of residues rich in carbon (C) is expected to improve the physical properties of soil [6] and enhance soil microbial activity [7] while reducing nitrogen (N) availability. The lower levels of N in soils amended with C-rich compost may promote plant nutrient stress, while C richness can affect the interactions between soil microorganisms and plants according to the C/N ratio, with positive implications for plant health [8], soil nutrient bioavailability [9], and soil structure [10]. Therefore, although maximum agronomic production requires high levels of available N, the use of C-rich residues and the consequent reduction in N availability, may improve plant quality. Organic fertilization, as practiced in organic farming systems, has produced crops with enhanced contents of antioxidants including phenolic compounds [11] with positive effects on human health [12], because the nutritional stress of crops stimulates the biosynthesis of these antioxidant compounds [13].

Due to its economic importance, tomato is one of the most analyzed foods in terms of varietal characterization and productivity. Plant susceptibility to pests and diseases can be affected by soil conditions [14,15,16] and plant composition [17]. In this context, several studies have been carried out in Barcelona province on tomato varieties to improve their resistance to pests and diseases and enhance productivity [18,19,20,21,22]. However, to the best of our knowledge, the effect of different types of organic fertilization on the production and quality of a range of traditional and high-yielding tomato varieties has not been previously studied in depth. Traditional tomato varieties are expected to be more suited to cultivation conditions with low or moderate N availability as compared with varieties that have been developed for high yields. We have chosen two widely used varieties and four local varieties with great culinary value. The aim of the present study was to investigate the sensitivity of such tomato varieties to organic fertilization, by testing three fertilization treatments with variable C and N input: N-rich, N-rich+C, and N-poor+C treatments. The examined parameters were production, plant nutritional status, and tomato quality. Regarding the nutritional quality of tomatoes, in this work, we test whether the oxidative stress of field tomato grown plants can be associated with, among others, an increase in phenolic compounds production. In view of the importance of the tomato crop, we investigate, for the first time, whether three organic fertilization treatments with different quantities of carbon and nitrogen affect the phenolic contents of contrasted tomato varieties.

## 2. Materials and Methods

### 2.1. Field Experiment

The study was carried out at the Can Gallina farm (Canet de Mar, Spain) located at 41°35′51.65″ N, 2°34′38.58″ E. The soil was a slightly basic (pH = 7.8) sandy clay with 2.3% organic C and 0.51% carbonates. Before setting up the experiment, the experimental area was fallowed for several years. Before tomato plantation, the area was milled to remove the grass cover and divided into nine plots.

### 2.2. Experimental Design

The study was performed with two varietal groups of tomato: ‘Montserrat’ and ‘Pebroter’. Three varieties were selected in each group: two traditional varieties and one high-yielding variety (Table 1).

Three different organic fertilizers with variable C/N ratios were used (Table 2): A commercial N-rich organic fertilizer and a K-rich mineral amendment were added to the N-rich plots. The N-rich+C plots were fertilized with a compost made from crushed woody plant residues (45%) and calf manure (55%). The N-poor+C plots were fertilized with a compost made from fine pruning residues (50%) and sheep manure (50%). The levels of total N in N-rich and N-rich+C fertilizer treatments were similar (Table 2), with lower amounts in the N-poor+C treatment. The lowest application of organic C was in the N-rich treatment. All fertilizers were spread on the soil surface and a dropwise irrigation system was installed. The soil surface was subsequently covered using biodegradable plastic (Mater-Bi) for weed control. Watering was carried out regularly according to plant needs.

Each treatment was replicated three times and randomly distributed within the nine plots of the experimental area. All tested tomato varieties were planted in each plot. The planting frame was 0.88 × 0.7 m, and in early May, two bushes of each variety were planted in the central part of each of the nine plots (Figure 1).

### 2.3. Leaf and Tomato Sampling and Analysis

Before tomato maturity in early July, mature leaf samples were taken from each tomato bush and bulked to one sample per treatment and variety. Before nutrient analysis, the leaves were oven-dried at 60 °C and finely ground using an automatic mortar grinder (RM 200, RETSCH, Haan, Germany). The C and N contents and isotope ratio (δ^13^C) in leaves were determined by using an isotope-ratio mass spectrometer (Flash 2000 HT, Thermo Fisher Scientific, Bremen, Germany). Nitrate was extracted from ground leaves suspended in deionized water (1/10 *w*/*w*) and the content was determined following the method of Cataldo et al. (1975) [23]. P, K, S, Ca, Mg, Na, Zn, Fe, Mo, and Cu contents in leaves were determined after digestion with HNO_3_ and HClO_4_ using inductively coupled plasma (simultaneous ICP-OES, Perkin Elmer Optima 8300, Waltham, MA 02451, USA) [24]. During the production period from mid-July to the first week of September, tomatoes were sampled once per week. The weights of tomatoes per treatment and variety were recorded and subsamples of tomatoes were frozen for the quality analyses. The most mature samples of each variety from all frozen samples were chosen according to the red color intensity: Stage 8 according to the Kleur Stadia (The Netherlandsd) tomato color chart [25]. Before further analyses, the Brix index was recorded for each tomato sample.

### 2.4. Phenolic Extraction and Determination

To perform the phenolic extraction, the method described by Rinaldi de Alvarenga et al. (2019) [26] was followed with few modifications. First, the tomatoes were ground until the samples were homogeneous. Then, 0.5 g was weighed for each sample and 5 mL of a dilution of ethanol and miliQ water (8:2 *v*/*v*) was added and the samples were sonicated for 5 min, vortexed for 30 s, and centrifuged at 4000 rpm for 20 min at 4 °C. The supernatant was transferred to another tube, and the extraction was repeated. After the second extraction, the supernatants from the two extractions were merged and evaporated with a vacuum evaporator (miVac DNA concentrator, Genevac LTD, Warminster, UK). Finally, all samples were reconstituted with 2 mL of miliQ water with 0.1% formic acid and stored at −80 °C until analyzed.

The phenolic compounds were identified and quantified by UHPLC-MS/MS, following the method described by Rinaldi de Alvarenga et al. (2019) [26]. An ACQUITY UPLC system equipped with a binary pump, autosampler, and oven from Waters (Milford, MA, USA) with a BEH C18 column (50 mm × 2.1 mm) i.d., 1.7 µm (Waters, Milford, MA, USA) was used. The injection volume was 10 µL, and the samples were maintained at 4 °C and the column at 30 °C. The mobile phase consisted of an A phase of acetonitrile (0.1% formic acid), and a B phase of water (0.1% formic acid). The gradient elution was: 0 min, 10% A; 0.5 min, 10% A; 1.5 min, 15% A; 2.0 min, 20% A; 2.5 min, 50% A; 3.0 min, 100% A; 3.5 min 100% A; and 4.5 min, 10% A. The flow rate applied was 400 µL/min.

An API 3000 triple quadrupole mass spectrometer (ABSciex, Framingham, MA, USA) coupled with a Turbo Ionspray source in negative ion mode was used for the MS/MS analysis. The settings of the Turbo Ionspray were the same as in Rinaldi de Alvarenga et al. (2019) [26]. The polyphenols were quantified using the multiple reaction monitoring mode (MRM), tracking the transition of parent ion and productions specific for each compound. The quantification was performed using the internal standard method, applying ethyl gallate as the internal standard, and generating calibration curves for each corresponding standard. The results were expressed as µg/g of the fraction.

### 2.5. Statistical Analysis

The data analysis was performed using SPSS 21.0 (SPSS Inc., Chicago, IL, USA), Statgraphics Centurion 18.1.13. (Statpoint Techonologies Inc., Warrenton, VA, USA) and Rstudio 3.6.3 (R Foundation for Statistical Computing, Vienna, Austria).

Data are presented as means and standard deviation. All data were tested for normality and homogeneity of variance. One-way analysis of variance (ANOVA) followed by Duncan’s multiple comparisons was used to evaluate the statistical differences in the concentrations of nutrients and phenolic compounds among the tomato varieties and fertilization treatments. A simple linear regression was applied to evaluate the association between total polyphenols and N-NO_3_ from tomato varieties.

Statistical tests were two-sided tests and statistical significance was set at *p*-value <0.05.

## 3. Results and Discussion

### 3.1. Effect of Fertilization on Tomato Yield

In organic agriculture, N is generally used as the base element to calculate the fertilization dose, as it is often the most limiting nutrient in agricultural soils [27]. In this study, three treatments based on C and N inputs were evaluated. Two treatments (N-rich and N-rich+C) contained the amount of N calculated as necessary for the crop and one treatment (N-poor+C) had a lower dose of N. Despite this variation, the tomato yields were quite uniform in all treatments, with no significant differences (Table 3). However, contradictory results have been observed. In a recent study, Carricondo-Martínez et al. (2022) [28] observed a reduced yield when using crop residues (C/N 15, yield 41.4 Tn/ha), composted goat manure (C/N 11, yield 41.0 Tn/ha), or composted vegetable waste (C/N 8.9, yield 43.2 Tn/ha) as compared with inorganic fertilizers used as a control (50 Tn/ha). While, in contrast, in other studies a higher production as compared with conventional mineral fertilization has been observed with the use of N-rich poultry manures or green manures [29,30]. Although the C/N ratio in our N-rich+C fertilizer was similar to that of the organic fertilizers used by Gatsios et al. (2021) [29], we did not observe changes in productivity in any tested variety, perhaps because of the high fertility of our experimental field. Similarly, Bénard et al. (2009) [31] reported that low N supply rates had little impact on commercial tomato fruit yield.

### 3.2. Productivity of the Different Varieties

Tomato production differed among the varieties, but mostly to a small extent (ranging from 18.73 to 20.54 Tn/ha). The exception was the high-yielding ‘Cornabel’, which stood out for its high productivity (32.24 Tn/ha). The Montserrat varietal group was the least productive, without differences between commercial and traditional varieties (Table 4). The yield was not affected by fertilization in any variety.

### 3.3. Effect of Fertilization on Mineral Nutrients in Plant Leaves

The contents of mineral elements in tomato leaves (Table 5) indicate nutrient availability in different agronomic practices [32,33]. In the present study, the major nutrients were not greatly affected by any of the treatments. As compared with the optimal reference levels [34], foliar N, Ca, and Mg were adequate with all fertilization treatments. On the one hand, regarding Mg, levels were increased by the N-rich treatment, probably because it was richer in this element (Table 2). The level of N in leaves suggests that the N content for each fertilizer was optimal. This is significant because a high application of this nutrient might shorten the useful life of the fruit by causing physiological alterations and even senescence [35] On the other hand, the levels of P and K were below the reference values in all cases; P did not differ among treatments, whereas significant increases in K were associated with the N-rich fertilization. An increase in Mo was observed in plants treated with the N-poor+C fertilizer, even though this treatment was Mo-poor. An essential micronutrient for plant growth, Mo is usually found in low concentrations in most plant tissues [36]. The availability of Mo depends on soil pH, the concentration of absorbing oxides, the extent of water drainage, and the organic compounds found in the soil colloids. In alkaline soils (as used in this study), Mo is more soluble and accessible to plants, while its availability decreases in acid soils [37]. Additionally, Na levels decreased with the N-poor+C treatment, and those of Zn increased with the N-rich treatment. No significant differences were observed in the S content, although a small increase was associated with the N-rich treatment. S is necessary for the normal metabolic processes of plants and is especially important for the biosynthesis of chlorophyll [36].

### 3.4. Mineral Nutrient Status in Tomato Varieties and Its Effect on δ^13^C and Nitrate Values

The concentrations of mineral nutrients in tomato leaves of the different varieties are shown in Table 6. No differences in N, P, K, and Mo were observed among the varieties. The highest Na content was observed in Montserrat Ple, followed by Corno Andino and Montserrat Fitó. Cu values were lowest in Cuban Pepper and highest in Montserrat Fitó (Figure 2A). N-NO_3_ accumulation was lowest in Corno Andino, and highest in Montserrat Ple (Figure 2B); storage of this highly soluble element can indicate an excess of N [38].

The concentrations of C were higher in the Montserrat than in the Pebroter varieties, although the values were significantly lower only in Corno Andino (Figure 3A). The concentration of C in the leaf was inversely related to the richness of mineral components.

Plants have adaption strategies to adverse conditions such as drought. A fundamental response of plants to water and heat stress is stomatal control of water loss. Closure of the stoma hinders the gaseous exchange of CO_2_, facilitating the accumulation of the δ^13^C isotope [39,40,41]. Thus, the higher the δ^13^C values, the greater the water stress. Following this premise, the plants that suffered the most water stress were the Montserrat varieties and those that suffered the least water stress were the Pebroter varieties (Figure 3B).

### 3.5. Phenolic Content

The six tomato varieties treated with three types of fertilization were analyzed for their phenolic contents. Fruit maturity according to ºBrix did not show statistically significant differences among the fertilization treatments (Appendix A). As expected, significant differences in phenolic levels were found between the varieties (Figure 4). In addition, the treatments had variable effects; the only significant increase was induced by the N-poor+C treatment in varieties of the Pebroter group (Figure 4), with specific phenolic compounds being particularly affected. In contrast, fertilization did not cause significant changes in the tomato phenolic content in the Monserrat group varieties, which were also more sensitive to water stress.

Regardless of the treatment, the highest amount of total phenolic compounds was found in Cuban Pepper tomatoes (Table 7). In this variety, phenolic levels were not affected by fertilization. Overall, the only significant changes observed in phenolic content were associated with the N-poor+C treatment, whose impact was positive in the Pebroter varieties Cornabel and Corno Andino, and negative in Can Duran, a variety particularly affected by water stress. Therefore, the results observed were different depending on the variety. A low N application combined with C (N-poor+C) favored the tomato phenolic contents in three of the studied varieties, as has been observed by Slimestad and Verheul (2005) [42], who described that the type of fertilization and other abiotic factors such as light affected the flavonoid content of tomatoes; however, in our case, had a negative impact on one variety, whereas two varieties were not affected by any treatment.

Regarding the levels of individual phenolic compounds (for more details of the individual phenolic compounds see in Appendix A), the response of each variety to each treatment becomes more complex (Table 7 and Table 8). Three compounds (naringenin, naringenin glucoside, and quercetin) were detected only in the Pebroter group, and therefore, may serve to discriminate it from the Montserrat group. Naringenin and naringenin derivatives are also reported to be absent in some Portuguese tomato varieties [43], whereas Slimestad et al. (2008) [44] detected minimal amounts of naringenin in nine different varieties of tomatoes. Naringenin was the most abundant phenolic compound in the three Pebroter varieties, and it was not significantly affected by fertilization, although the differences in concentrations among the varieties became significant after the N-poor+C fertilization treatment: naringenin levels were highest in Cuban Pepper (59.08 mg/kg), followed by Corno Andino (36.19 mg/kg), and Cornabel (10.94 mg/kg). Regarding naringenin glucoside and quercetin, fertilization had different effects in each Pebroter variety. In Cuban Pepper, naringenin glucoside levels were significantly higher with N-rich+C fertilization treatment as compared with N-rich and N-poor+C fertilization treatment, whereas, in Cornabel, they were significantly lower with N-rich+C versus N-rich treatment. On the other hand, in Cuban Pepper, quercetin was significantly higher with N-rich+C treatment as compared with N-poor+C treatment, whereas the reverse was found in Corno Andino. The second highest polyphenol was rutin in the Pebroter group. In contrast, Cruz-Carrión et al. (2022) [45] observed rutin as the highest flavonoid in local tomatoes from Tarragona. In addition, Gomez-Romero et al. (2010) [46] reported differences in the content of phenolic compounds in two different varieties of tomatoes, and Slimestad et al. (2008) [44] reported differences in the contents of flavonoids among nine tomato varieties analyzed. In the latter study, Naringenin chalcone was the main flavonoid, followed by rutin, although, in some varieties, quercetin 3-(2″-apiosyl-6″-rhamnosyl-glucoside and phloretin 3′,5′-di-*C*-glucoside were present at levels similar to rutin. The Montserrat cultivars tested in this study showed a distinct polyphenolic composition as compared with other varieties as they showed the highest levels of chlorogenic acid and coumaric acid glucoside. Similar to what we observed in Montserrat varieties, Benard et al. (2009) [31] did not observe a general increase in phenolic compounds in response to a decrease in N supply in tomato crops, however, in this latter study, some compounds did have an increasing trend. Stewart et al. (2001) [47] also found no effects on flavonoid content in tomatoes grown under low N. The C and N richness of organic fertilization had different effects on each phenolic compound in each variety studied. For example, the Cornabel variety responded most favorably to the N-poor+C treatment, which enhanced the content of almost all the phenolic compounds identified in this variety. Yet, the same treatment had a negative effect on Can Duran, reducing the concentrations of almost all phenolic compounds, which, in contrast, were enhanced by N-rich and N-rich+C fertilization. The effects of the different fertilization treatments on Cuban Pepper were largely similar, although the levels of apigenin glucoside, o-coumaric acid, p-coumaric acid, ferulic glucoside, and caffeic hexoside were increased more by the N-poor+C treatment, and those of caffeic acid, protocatechuic, quercetin, rutin, and naringenin glucoside were increased more by N-rich+C fertilization. The phenolic content of Corno Andino tomatoes was improved the most by the N-rich+C treatment, which significantly increased levels of chlorogenic, m-coumaric, o-coumaric and p-coumaric acid, quercetin, coumaric glucoside, and ferulic glucoside. Divergent effects were observed in Montserrat Fitó, although, overall, N-poor+C seemed to be the most favorable treatment, significantly increasing levels of 4-hydroxybenzoic acid, m-coumaric acid, o-coumaric acid, and coumaric glucoside; otherwise, increases in apigenin glucoside were associated with both high and low N+C, homovanillic acid glucoside, and protocatechuic acid with N-rich+C treatment, and rutin with both N-rich+C and N-rich treatments. Finally, Montserrat Ple seemed to be barely affected by fertilization, although 4-hydroxybenzoic, apigenin glucoside, homovanillic acid glucoside, and caffeic hexoside were favored by N-poor+C treatment, caffeic acid and p-coumaric acid by N-rich+C treatment, and o-coumaric acid by both low and high N+C treatments. In summary, no clear trend was observed in the effect of organic fertilization on the levels of individual phenolic compounds, which differed in each variety, although the N-poor+C treatment was generally the most favorable (Appendix A).

According to the literature, a reduced N supply generates an accumulation of phenolic compounds as a response to the abiotic stress of N limitation [35]. Thus, Bénard et al. (2011) [48] reported that temporary deprivation of N increased the phenolic content in tomato plant leaves. Similar behavior was observed in Cornabel, Corno Andino, and Montserrat Fitó varieties that had the highest phenolic content with N-poor+C treatment. However, the opposite behavior was observed in Montserrat Can Duran. Nevertheless, it is clearly observed that cultivars, depending on the variety, accumulate different profiles and proportions of antioxidant molecules, such as polyphenols [49].

### 3.6. Plant Nutritional Status, Water Stress Sensitivity, and Oxidative Stress Regulation

Nutrient shortages and water stress can both limit photosynthesis, and thus, induce oxidative stress that may enhance antioxidant production in plants [13,50]. Since leaf N-NO_3_ content in plants growing under similar conditions can be used as an indicator of the luxury consumption of N [38], we used N-NO_3_ as a covariable to analyze the use of N in each varietal group and fertilization treatment. The results showed significant differences only in the Pebroter varieties, with interactions between fertilizer and leaf N-NO_3_ (*p* = 0.023) and with each variety (*p* = 0.027). In this case, total polyphenol content increased with N-NO_3_ only with N-poor+C fertilization, and not the N-rich treatments (N-rich and N-rich+C, Figure 5A). In contrast, with N-rich+C fertilization, a lower content of N-NO_3_ in the leaves was associated with a higher total polyphenol content, suggesting that polyphenol production in Pebroter tomatoes may be limited by N only in N-poor environments. This agrees with Bénard et al. (2009) [31], who reported increases in some phenolic compounds in tomatoes grown in soil supplied with low quantities of N (4 mM). Therefore, in our experiment, the highest levels of phenolics were found in Pebroter tomato varieties cultivated with N-poor and C-rich fertilization, and their synthesis was limited by N availability. In contrast, in N-rich environments, overall polyphenol production was lower and not limited by N. However, this trend was not observed in the Montserrat group, whose varieties showed high sensitivity to water stress (Figure 5B).

In general, plants respond to water stress by increasing the synthesis of antioxidant compounds such as phenolic compounds and terpenes [51,52,53,54,55]. Sánchez-Rodríguez et al. (2011) [56] reported an increase in phenolic compounds in cherry tomatoes exposed to water stress. Another study observed that, while the total flavonoid content of tomatoes was not influenced by water stress, the total phenolic content increased [51]. Similar observations in other crops such as ‘merlot’ cultivar grapes have been reported [57]. This contrasts with the results of the present study, in which the Montserrat varieties suffered the most water stress yet produced the fewest total phenolic compounds; moreover, the phenolic content decreased with increasing water stress (r = −0.412, *p* = 0.033). These conflicting results might be justified either by considering that the water stress of our tomato varieties was not induced, and therefore, moderate as compared with other studies, or by the fact that oxidative stress regulation in Montserrat varieties may be less related to total polyphenol content. Indeed, water stress activates a network of antioxidant defences to prevent cellular damage caused by photo-oxidative stress [58]. In addition, decreased N availability in N-poor+C treatment improves phenol content only in the water stress less sensitive Pebroter varietal group. The oxidative stress due to reduced N supply seems to affect phenol production in both traditional low yielding and high yielding varieties of the Pebroter varietal group. Similar results in total polyphenol content due to low nutrient availability have been found in other plant species [59,60].

## 4. Conclusions

In the present research, although none of the tested fertilizer tratments influenced the productivity of the studied varieties, N-poor+C fertilization increased the tomato phenolic contents in the Pebroter varieties. With respect to the Montserrat varieties and treatments with higher N availability, indeed, the phenol contents of the Montserrat varieties were not sensitive to the fertilization treatments and decreased with water stress. Moreover, naringenin, the most abundant individual phenolic compound in the Pebroter varietal group, was not present in the Montserrat varietal group. In summary, regarding total phenolic compound synthesis, reduced N in organic fertilization increased the quality of tomato in the Pebroter varietal group while it did not influence the yield in any of the tested varieties. Reducing N in organic amendments may, therefore, be a promising approach for increasing the polyphenolic antioxidant compounds in some varieties of tomatoes, which may have benefits for human health.

## Figures and Tables

**Figure 1 antioxidants-11-02127-f001:**
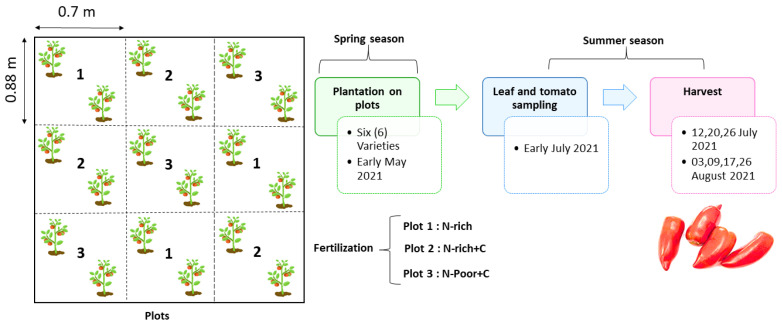
Field experimental layout of the application of different doses of N and C in the cultivation of six tomato varieties.

**Figure 2 antioxidants-11-02127-f002:**
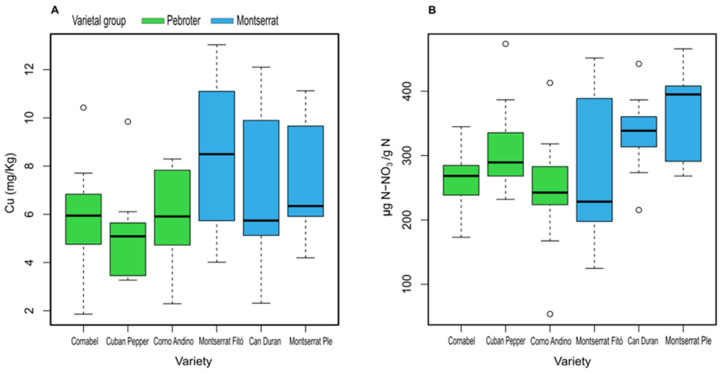
Concentrations of Cu (**A**) and N-NO_3_ (**B**) in leaves of tomato varieties.

**Figure 3 antioxidants-11-02127-f003:**
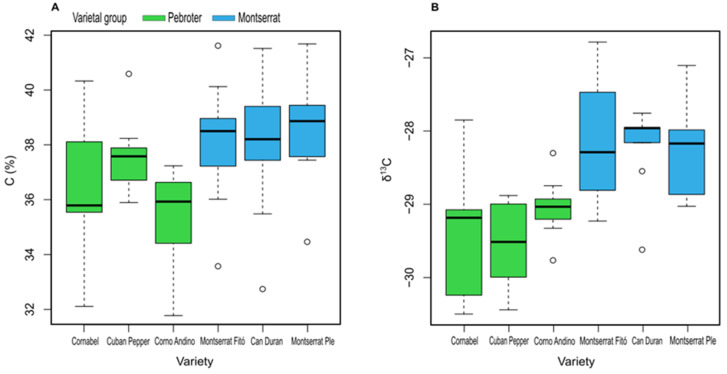
C (**A**) and δ^13^C (**B**) in the leaves of tomato varieties.

**Figure 4 antioxidants-11-02127-f004:**
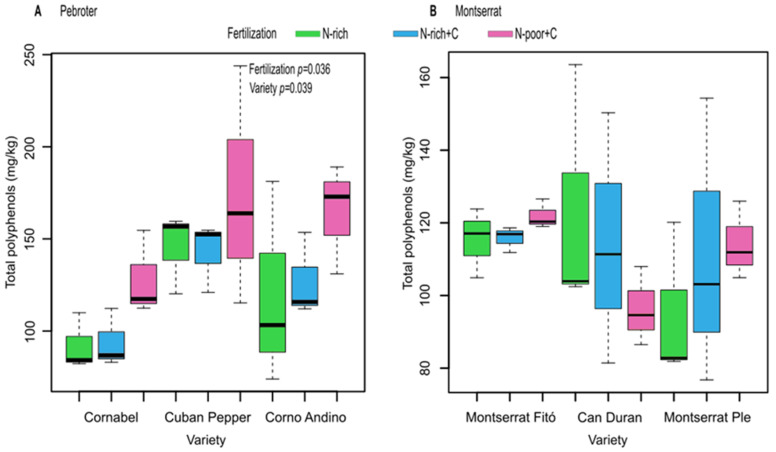
Total polyphenol content in tomato varieties, (**A**) Pebroter varietal group, (**B**) Montserrat varietal group.

**Figure 5 antioxidants-11-02127-f005:**
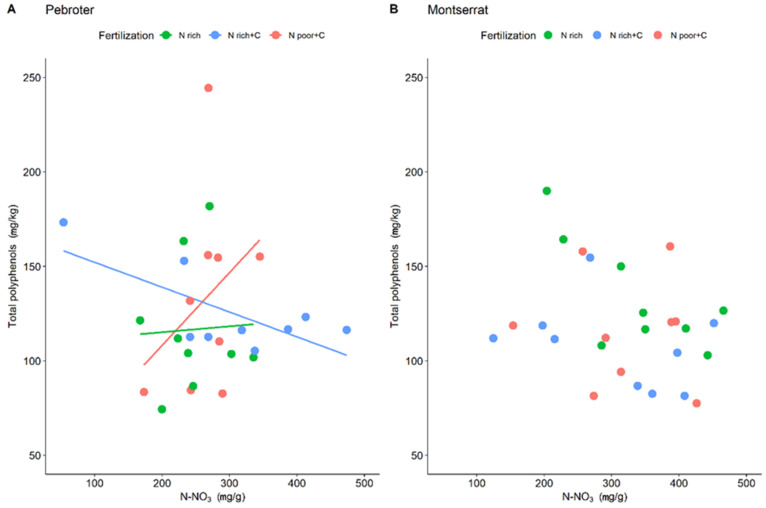
Effect of fertilization and nitrates (N-NO_3_) on total phenolic content in Pebroter (**A**) and Montserrat (**B**) varietal groups.

**Table 1 antioxidants-11-02127-t001:** Selected tomato varieties.

	Varietal Group	Variety	Type
1	Pebroter	Cornabel	High-yielding
2	Pebroter	Cuban Pepper	Traditional
3	Pebroter	Corno Andino	Traditional
4	Montserrat	Montserrat Fitó	High-yielding
5	Montserrat	Can Duran	Traditional
6	Montserrat	Montserrat Ple	Traditional

**Table 2 antioxidants-11-02127-t002:** Chemical composition of each fertilizer and contributions of C and nutrients in each treatment.

	N-rich Fertilizer C/N 4	N-rich+C Fertilizer C/N 10	N-poor+C Fertilizer C/N 20
Dry weight (%)	-	53.1	58.7
C (%)	40	29.5	14.4
N (%)	10	2.7	0.83
C/N	4	10.9	17.3
N-NH_4_^+^ (%)	-	0.5	0.1
pH	-	7.8	8.94
CE (µS/cm)	-	6010	770
C (kg/ha)	1000	2741	1775
N (kg/ha)	250	251	102
N-NH4 (kg/ha)	-	46	12
P (kg/ha)	-	84	37
K (kg/ha)	750	158	136
Mg (kg/ha)	180	74	111
S (kg/ha)	510	12	12
Dose (kg/m2)	0.25/0.3	1.7	2.1

**Table 3 antioxidants-11-02127-t003:** Production according to the type of organic fertilization.

	N-rich Fertilizer C/N 4	N-rich+C Fertilizer C/N 10	N-poor+C Fertilizer C/N 20
Production (Tn/ha)	22.05 ± 7.76	23.59 ± 10.59	18.37 ± 7.58

Data are the mean ± standard deviation.

**Table 4 antioxidants-11-02127-t004:** Productivity of tomato varieties from the organic fertilization test.

	Varietal Group	Variety	Production (Tn/ha)
1	Pebroter	Cornabel	32.24 ± 11.57 a
2	Pebroter	Cuban Pepper	20.54 ± 5.51 b
3	Pebroter	Corno Andino	19.04 ± 5.65 b
4	Montserrat	Montserrat Fitó	18.06 ± 5.69 b
5	Montserrat	Can Duran	19.40 ± 8.20 b
6	Montserrat	Montserrat Ple	18.73 ± 7.80 b

Data are shown as mean ± standard deviation (*n* = 3) for each varietal group. Different letters represent significant differences in production observed between varietal groups (*p* < 0.05).

**Table 5 antioxidants-11-02127-t005:** Leaf mineral nutrients at the beginning of tomato curdling in plants treated with organic fertilization. The reference for optimal levels is from Villar and Villar (2016) [34].

Parameter	Optimal Level	N-rich Fertilizer C/N 4	N-rich+C Fertilizer C/N 10	N-poor+C Fertilizer C/N 20
N (%)	2.9–4	3.77 ± 0.28	3.69 ± 0.42	3.73 ± 0.56
P (%)	0.3–0.75	0.22 ± 0.05	0.19 ± 0.07	0.22 ± 0.08
K (%)	2.1–4.7	1.42 ± 0.74 b	0.91 ± 0.47 a	1.01 ± 0.49 ab
Ca (%)	2.6–7.0	3.76 ± 1.28	3.06 ± 1.18	3.84 ± 1.39
Mg (%)	0.3–0.9	0.84 ± 0.23 a	0.60 ± 0.19 b	0.66 ± 0.20 ab
Na (mg/Kg)		744.65 ± 597.28 ab	807.68 ± 412.68 a	492.51 ± 195.72 b
Zn (mg/Kg)		22.61 ± 9.97 a	16.30 ±7.47 b	17.16 ± 6.60 ab
S (%)		1.38 ± 0.57	1.13 ± 0.48	1.19 ± 0.52
Fe (%)		0.03 ± 0.008	0.03 ± 0.010	0.03 ± 0.008
Mo (mg/kg)		6.05 ± 11.56	6.56 ± 5.71	8.0 ± 6.06
C (%)		37.12 ± 2.13	37.11 ± 2.88	37.65 ± 2.01

Data are shown as mean ± standard deviation. Different letters represent significant differences in nutrients among fertilization treatments (*p* < 0.05).

**Table 6 antioxidants-11-02127-t006:** Leaf mineral nutrients of tomato varieties.

Parameter	Cornabel	Cuban Pepper	Corno Andino	Montserrat Fitó	Can Duran	Montserrat Ple
N (%)	3.59 ± 0.63	4.01 ± 0.39	3.72 ± 0.32	3.74 ± 0.34	3.67 ± 0.33	3.66 ± 0.48
P (%)	0.21 ± 0.09	0.23 ± 0.06	0.23 ± 0.07	0.22 ± 0.06	0.20 ± 0.08	0.19 ± 0.03
K (%)	1.04 ± 0.64	1.22 ± 0.62	1.18 ± 0.76	1.13 ± 0.63	0.89 ± 0.49	1.22 ± 0.64
Na (mg/Kg)	583.79 ± 483.2	540.79 ± 175.07	783.13 ± 295.98	751.18 ± 455.44	501.24 ± 298.83	929.53 ± 717.34
Mo (mg/kg)	4.66 ± 3.13	5.72 ± 6.52	4.84 ± 1.91	8.23 ± 7.09	5.84 ± 4.84	11.96 ± 16.20
C (%)	36.50 ± 2.52 ab	37.55 ± 1.41 b	35.28 ± 2.0 a	38.01 ± 2.32 b	37.98 ± 2.57 b	38.44 ± 1.99 b
Cu (mg/Kg)	5.89 ± 2.38 ab	5.17 ± 2.04 a	6.03 ± 2.02 ab	8.66 ± 3.03 b	7.20 ± 3.38 ab	7.42 ± 2.37 ab
N-NO_3_ (µg/g)	262.06 ± 56.92 ab	309.91 ± 78.53 ab	243.17 ±98.74 a	268.46 ± 118.72 ab	332.67 ± 65.22 ab	364.82 ± 70.07 b

Data are shown as mean ± standard deviation. Different letters represent significant differences in nutrients among varieties (*p* < 0.05).

**Table 7 antioxidants-11-02127-t007:** Concentrations of the phenolic compounds (mg/kg) in the Pebroter tomatoes according to the fertilization used.

Phenolic Compound	Cornabel			Cuban Pepper			Corno Andino		
	N-rich	N-rich+C	N-poor+C	N-rich	N-rich+C	N-poor+C	N-rich	N-rich+C	N-poor+C
4-Hydroxybenzoic acid	0.41 ± 0.11	0.37 ± 0.03	0.56 ± 0.07	0.46 ± 0.03	0.59 ± 0.12	0.61 ± 0.09	0.52 ± 0.02	0.46 ± 0.06	0.57 ± 0.16
Gallic acid	12.08 ± 0.62	12.87 ± 0.67	12.61 ± 0.91	12.59 ± 0.63	12.65 ± 0.91	12.57 ± 0.65	12.24 ± 0.44	12.24 ± 0.91	12.63 ± 0.58
Caffeic acid	0.71 ± 0.63	1.70 ± 0.32	1.71 ± 0.50	1.48 ± 0.27	2.16 ± 0.48	1.21 ± 0.82	2.54 ± 0.24	2.76 ± 0.86	2.67 ± 1.33
Caffeic acid hexoside	3.66 ± 1.37	6.11 ± 2.43	6.55 ± 2.62	4.03 ± 1.03	5.38 ± 1.01	6.10 ± 2.60	5.86 ± 2.63	5.87 ± 1.17	5.68 ± 1.91
Chlorogenic acid	5.47 ± 4.57	14.08 ± 4.10	22.38 ± 6.41	13.99 ± 3.07	13.36 ± 3.96	19.07 ± 12.60	10.85 ± 5.09	13.38 ± 3.45	19.01 ± 2.61
Neochlorogenic acid	0.05 ± 0.00	0.09 ± 0.02	0.11 ± 0.03	0.09 ± 0.02	0.09 ± 0.01	0.11 ± 0.02	0.09 ± 0.02	0.12 ± 0.04	0.07 ± 0.04
Cryptochlorogenic acid	1.20 ± 0.12	1.40 ± 0.31	1.75 ± 0.41	1.18 ± 0.20	1.36 ± 0.53	1.21 ± 0.65	1.49 ± 0.74	1.85 ± 0.46	1.83 ± 0.39
Protocatechuic acid	1.37 ± 1.00	0.68 ± 0.29	1.27 ± 0.44	0.64 ± 0.23	0.70 ± 0.14	0.49 ± 0.14	1.17 ± 0.25	1.07 ± 0.49	1.14 ± 0.30
Dicaffeyolquinic acid	0.32 ± 0.24	0.91 ± 0.42	0.91 ± 0.23	1.10 ± 0.26	0.89 ± 0.38	1.05 ± 0.71	1.01 ± 0.49	1.12 ± 0.11	1.07 ± 0.08
*m*-Coumaric acid	2.53 ± 0.63	3.78 ± 1.02	5.38 ± 0.39	6.83 ± 1.86	7.05 ± 4.00	6.65 ± 1.20	2.87 ± 0.13	3.79 ± 1.24	5.96 ± 1.43
*o*-Coumaric acid	3.36 ± 1.47	4.48 ± 1.30	5.24 ± 3.01	3.54 ± 0.88	4.38 ± 0.79	7.58 ± 2.79	5.26 ± 0.62	6.34 ± 2.89	10.78 ± 3.94
*p*-Coumaric acid	0.71 ± 0.17	0.48 ± 0.11	0.94 ± 0.23	1.08 ± 0.21	0.99 ± 0.17	1.23 ± 0.17	1.15 ± 0.44	1.26 ± 0.75	2.59 ± 1.36
Coumaric acid glucoside	9.03 ± 2.84	12.82 ± 2.91	16.98 ± 3.66	21.73 ± 5.72	22.05 ± 9.69	21.80 ± 1.75	11.50 ± 1.16	15.33 ± 1.19	23.97 ± 6.39
Homovanillic acid glucoside	0.25 ± 0.03	nd	0.28 ± 0.02	0.23 ± 0.01	0.27 ± 0.04	0.29 ± 0.05	0.24 ± 0.03	0.24 ± 0.00	nd
Ferulic acid glucoside	0.35 ± 0.06	0.49 ± 0.16	0.70 ± 0.09	0.47 ±0.04	0.55 ± 0.16	0.65 ± 0.27	0.48 ± 0.19	0.51 ± 0.06	0.73 ± 0.05
Rutin	14.05 ± 5.37	14.95 ± 2.81	28.53 ± 7.70	24.56 ± 3.29	29.29 ± 4.72	20.85 ± 3.74	18.98 ± 10.11	20.71 ± 5.31	15.80 ± 7.87
Quercetin	1.56 ± 0.12	1.58 ± 0.13	1.62 ± 0.15	1.69 ± 0.14	1.97 ± 0.32	1.59 ± 0.13	1.82 ± 0.41	1.76 ± 0.31	2.46 ± 1.05
Naringenin	20.34 ± 16.51	8.79 ± 9.01	10.94 ± 9.44	41.83 ± 41.73	23.84 ± 19.80	59.08 ± 44.40	31.16 ± 24.84	23.99 ± 14.96	36.19 ± 24.14
Naringenin glucoside	11.93 ± 7.35	5.93 ± 4.38	9.45 ± 2.05	6.78 ± 1.48	11.99 ± 6.99	7.22 ± 4.03	13.82 ± 9.56	11.91 ± 4.99	17.33 ± 12.81
Apigenin glucoside	1.34 ± 0.32	1.70 ± 0.48	1.53 ± 0.39	1.60 ± 0.37	1.63 ± 0.14	2.05 ± 0.25	1.73 ± 0.29	2.28 ± 0.40	2.21 ± 0.67
Total phenolics	90.72 ± 15.54	93.21 ± 16.01	129.44 ± 23.21	145.9 ± 22.05	141.19 ± 19.00	171.41 ± 65.33	124.78 ± 55.78	126.99 ± 23.10	162.78 ± 30.12

Data are the mean ± deviation standard. nd refer to non detectable.

**Table 8 antioxidants-11-02127-t008:** Concentrations of the phenolic compounds (mg/kg) in the Montserrat tomatoes according to the fertilization used.

Phenolic Compound	Montserrrat Fitó			Can Duran			Montserrat Ple		
	N-rich	N-rich+C	N-poor+C	N-rich	N-rich+C	N-poor+C	N-rich	N-rich+C	N-poor+C
4-Hydroxybenzoic acid	0.44 ± 0.10	0.56 ± 0.13	0.62 ± 0.14	0.63 ± 0.09	0.69 ± 0.23	0.60 ± 0.21	0.68 ± 0.11	0.55 ± 0.15	0.99 ± 0.23
Gallic acid	12.40 ± 1.56	12.68 ± 0.76	12.47 ± 0.58	12.31 ± 0.53	12.92 ± 0.72	12.82 ± 1.05	12.13 ± 0.83	12.91 ± 0.43	12.88 ± 0.84
Caffeic acid	2.29 ± 0.74	2.28 ± 0.51	3.38 ± 1.83	2.26 ± 0.42	2.06 ± 0.40	1.71 ± 0.81	1.59 ± 0.95	2.56 ± 0.75	1.63 ± 0.93
Caffeic acid hexoside	10.40 ± 6.62	11.12 ± 1.42	13.11 ± 1.25	11.95 ± 1.40	15.05 ± 5.87	7.78 ± 2.36	9.94 ± 2.09	10.80 ± 2.92	14.98 ± 3.86
Chlorogenic acid	22.09 ± 6.34	17.62 ± 2.46	20.45 ± 7.86	20.94 ± 3.32	18.26 ± 1.47	14.57 ± 4.01	15.47 ± 3.10	22.55 ± 15.37	14.01 ± 9.33
Neochlorogenic acid	0.12 ± 0.04	0.13 ± 0.06	0.12 ± 0.05	0.13 ± 0.03	0.12 ± 0.04	0.10 ± 0.06	0.14 ± 0.06	0.13 ± 0.05	0.19 ± 0.16
Cryptochlorogenic acid	1.68 ± 1.18	1.47 ± 0.74	2.19 ± 0.65	2.40 ± 0.42	1.98 ± 0.27	1.84 ± 0.39	1.74 ± 0.37	1.36 ± 0.34	1.40 ± 0.83
Protocatechuic acid	0.62 ± 0.18	1.05 ± 0.48	0.94 ± 0.24	0.64 ± 0.17	0.67 ± 0.12	0.73 ± 0.38	0.74 ± 0.30	0.84 ± 0.40	0.92 ± 0.28
Dicaffeyolquinic acid	1.30 ± 0.51	1.09 ± 0.74	1.10 ± 0.42	1.09 ± 0.71	0.90 ± 0.32	0.60 ± 0.14	0.82 ± 0.45	1.25 ± 0.69	1.07 ± 0.68
*m*-Coumaric acid	5.92 ± 1.71	5.36 ± 0.83	7.82 ± 2.12	5.97 ± 0.39	4.89 ± 1.88	5.95 ± 0.82	3.99 ± 0.92	3.90 ± 1.65	5.21 ± 1.78
*o*-Coumaric acid	11.44 ± 4.16	15.98 ± 0.89	16.80 ± 4.92	12.44 ± 1.45	17.55 ± 7.23	11.86 ± 2.40	9.97 ± 2.80	13.07 ± 1.72	14.44 ± 2.41
*p*-Coumaric acid	1.39 ± 0.68	2.23 ± 1.19	2.46 ± 1.86	1.63 ± 0.41	1.61 ± 0.22	1.79 ± 0.58	1.18 ± 0.11	1.57 ± 0.34	1.02 ± 0.18
Coumaric acid glucoside	26.21 ± 8.06	27.67 ± 2.58	32.66 ± 5.05	26.60 ± 1.98	27.80 ± 10.29	22.59 ± 4.01	18.71 ± 5.08	20.77 ± 6.20	23.82 ± 4.63
Homovanillic acid glucoside	0.43 ± 0.10	0.52 ± 0.13	0.40 ± 0.11	0.53 ± 0.06	0.57 ± 0.22	0.32 ± 0.07	0.44 ± 0.04	0.41 ± 0.08	0.60 ± 0.18
Ferulic acid glucoside	0.89 ± 0.56	1.02 ± 0.48	1.39 ± 0.64	1.23 ± 0.31	1.48 ± 0.38	1.37 ± 0.39	0.86 ± 0.20	0.89 ± 0.25	0.74 ± 0.20
Rutin	9.63 ± 7.17	9.46 ± 6.75	0.75 ± 0.29	3.47 ± 4.69	0.58 ± 0.12	2.87 ± 5.98	12.23 ± 6.91	13.45 ± 10.51	13.64 ± 2.51
Quercetin	nd	nd	nd	nd	nd	nd	nd	nd	nd
Naringenin	nd	nd	nd	nd	nd	nd	nd	nd	nd
Naringenin glucoside	nd	nd	nd	nd	nd	nd	nd	nd	nd
Apigenin glucoside	3.18 ± 0.75	4.92 ± 1.10	4.07 ± 0.53	2.22 ± 0.90	3.76 ± 1.00	2.01 ± 0.43	3.36 ± 0.83	3.12 ± 1.08	5.60 ± 1.21
Total phenolics	110.43 ± 9.59	115.16 ± 3.52	120.73 ± 4.05	123.31 ± 34.86	114.37 ± 34.54	96.36 ± 10.85	94.95 ± 21.83	110.08 ± 39.44	112.54 ± 10.73

Data are the mean ± deviation standard. nd refer to non detectable.

## Data Availability

Data is contained within the article or Appendix A.

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
