# Peer review of "The Effects of Differentiated Organic Fertilization on Tomato Production and Phenolic Content in Traditional and High-Yielding Varieties"

_antioxidants, 2022, doi:10.3390/antiox11112127_

Round 1

Reviewer 1 Report

Comments on the paper entitled „Effect of differentiated organic fertilization on tomato production and phenolic content in traditional and high-yielding varieties”Manuscript numer: antioxidants-1952141

The work concerns the assessment of the content of polyphenols in various tomato cultivars under the influence of different fertilizing components, i.e. with different N and C content, and the influence of fertilization on productivity of the studied varieties and on mineral nutrients in plant leaves.

The manuscript  generally is well written. However, in the Results and Discussion section, the Authors did not relate their research results sufficiently to the studies of other authors who used similar fertilization,  tomato varietes and studied the content of phenols and minerals. This applies in particular to chapter 3.3. and 3.5. I propose to complete the Discussion section.

Line 89: please list, what were organic fertilizers and please list their components, not only the levels of the C and N and nutrients and ratios C/N.

Author Response

Point 1: The work concerns the assessment of the content of polyphenols in various tomato cultivars under the influence of different fertilizing components, i.e. with different N and C content, and the influence of fertilization on productivity of the studied varieties and on mineral nutrients in plant leaves. 

The manuscript generally is well written. However, in the Results and Discussion section, the Authors did not relate their research results sufficiently to the studies of other authors who used similar fertilization, tomato varietes and studied the content of phenols and minerals. This applies in particular to chapter 3.3. and 3.5. I propose to complete the Discussion section. 

Response 1: Thanks for the suggestion. By adding new related references, we have tried to improve the discussion of the points you have highlighted. In section3.5 we compare the content of individual phenols in tomato varieties with those obtained by other authors, and we remark that as, other authors have suggested, the content of phenols varies according to the type of fertilization (Lines 261-264; 274-275, and 285-299). However, in section3.3, the nutritional condition of the plant employing organic fertilizer, as far as we know, has not been reported by previously. 

Nevertheless, we point out that the nitrogen content given to each fertilizer under study was optimal because it did not accumulate as nitrate in the tomato leaves, which is beneficial for the health of the plant (Lines 202-204).  

Point 2: Line 89: please list, what were organic fertilizers and please list their components, not only the levels of the C and N and nutrients and ratios C/N. 

Response 2: According to the suggestion of the reviewer, we have added the components of each organic fertilizer that we used in this experiment, with their respective percentages (lines 92-96) 

Reviewer 2 Report

The main question addressed by the research is effect of organic fertilization on phenolic content of tomato. The topic is not original, however I consider it to be interesting and addressing an important issue. It provides relevant information for researchers and producers working with tomato. The use of known commercial varieties of tomato, which are the ones more consumed, perhaps functioning as controls.

The presentation can be improved as well as the introduction.

The introduction and general presentation can be improved. The work would benefit from the use of one the most commercially used tomato varieties.

Author Response

Point 1: The main question addressed by the research is effect of organic fertilization on phenolic content of tomato. The topic is not original, however I consider it to be interesting and addressing an important issue. It provides relevant information for researchers and producers working with tomato. The use of known commercial varieties of tomato, which are the ones more consumed, perhaps functioning as controls. 

The presentation can be improved as well as the introduction. 

The introduction and general presentation can be improved. The work would benefit from the use of one the most commercially used tomato varieties. 

Response 1: We have added the varieties of tomatoes that we have selected, two commercial varieties and four traditional varieties, which have been chosen due to their great culinary value (Lines 71-71 and 78-80). On the other hand, the significance of tomato crops is discussed in the introduction, along with how adding carbon-rich residues to the soil is expected to reduce the surplus of nitrogen and as consequence, can increase the biosynthesis of antioxidant compounds. The importance of our experiments, in which the relationship between fertilizer C/N ratio and quality in several tomato varieties was evaluated for the first time, is explained in the introduction. In the introduction we have added the following sentence: “In view of the importance of the tomato crop, we investigate for the first time whether three organic fertilization regimes with different quantities of carbon and nitrogen affect the phenolic content of contrasted tomato varieties.” 
